# Watch out! Motion is Blurring the Vision of Your Deep Neural Networks

**Qing Guo**[1]  **Felix Juefei-Xu**[2]  **Xiaofei Xie**[1*]  **Lei Ma**[3*]

**Jian Wang**[1]  **Bing Yu**[3]  **Wei Feng**[4]  **Yang Liu**[1]

[1]Nanyang Technological University, Singapore  [2]Alibaba Group, USA
[3]Kyushu University, Japan  [4]Tianjin University, China

## Abstract

The state-of-the-art deep neural networks (DNNs) are vulnerable to adversarial examples with additive random noise-like perturbations. While such examples are hardly found in the physical world, the image blurring effect caused by object motion, on the other hand, commonly occurs in practice, making the study of which greatly important especially for the widely adopted real-time image processing tasks (*e.g.*, object detection, tracking). In this paper, we initiate the first step to comprehensively investigate the potential hazards of blur effect for DNN, caused by object motion. We propose a novel adversarial attack method that can generate visually natural motion-blurred adversarial examples, named motion-based adversarial blur attack (ABBA). To this end, we first formulate the kernel-prediction-based attack where an input image is convolved with kernels in a pixel-wise way, and the misclassification capability is achieved by tuning the kernel weights. To generate visually more natural and plausible examples, we further propose the saliency-regularized adversarial kernel prediction, where the salient region serves as a moving object, and the predicted kernel is regularized to achieve visual effects that are natural. Besides, the attack is further enhanced by adaptively tuning the translations of object and background. A comprehensive evaluation on the NeurIPS'17 adversarial competition dataset demonstrates the effectiveness of ABBA by considering various kernel sizes, translations, and regions. The in-depth study further confirms that our method shows more effective penetrating capability to the state-of-the-art GAN-based deblurring mechanisms compared with other blurring methods. We release the code to https://github.com/tsingqguo/ABBA.

## 1 Introduction

Deep neural networks (DNN) have been widely applied in various vision perception tasks (*e.g.*, object recognition, segmentation, scene understanding), permeating many aspects of our daily life, such as autonomous driving, robotics, video surveillance, photo taking, *etc*. However, the state-of-the-art DNNs are still vulnerable to adversarial examples. Extensive previous works are proposed (*e.g.*, FGSM [1], BIM [2], MI-FGSM [3], C&W [4]), to mislead the DNN through additive noise perturbations that could be obtained by optimizing the adversarial objectives. To be imperceptible to human, $L_\mathrm{p}$-norm plays an important role in such attacks, confining the perturbation noise to be small. However, the random noise-like perturbation often does not pose imminent threats to the camera systems, which does not usually occur in natural environment. Thus, some recent attempts [2] were made to physically fashion adversarial examples such as by putting up stickers or printed patterns

on the physical stop sign, *etc*. Again, these artifacts are often intentionally prepared for adversarial attacks, which are not 'naturally' found in the real-world environment either.

While the blurring effect caused by object motion commonly occurs in practical image perception systems, the potential hazards of motion blur effect to the DNN are largely untouched so far. Motion blur naturally happens during the exposure time of image capturing. When an object moves at a relatively high speed, all information of the object during the image capture process is integrated, constituting a blur-like image along a relative moving direction. Compared with other kinds of image blur (*e.g.*, defocus blur caused by using unsuitable camera focus), motion blur is directly related to the motion of object and camera, whose effect cannot be easily removed by adjusting the camera's setting. As a result, motion blur almost coexists with the camera and potentially posts serious effects on DNN perception-based systems. However, up to present, there are limited studies discuss how motion blur affects the DNN perception tasks. It is not even clear whether and what kinds of motion blur can systematically mislead a DNN.

In this paper, we initiate the first step to comprehensively investigate the blur effects to DNNs from the adversarial attack perspective, where systematic motion blur-based adversarial example discovery would be an important step towards further DNN enhancement. In particular, we propose a new type of adversarial attack, termed motion-based adversarial blur attack (ABBA), which can generate visually natural and plausible motion-blurred adversarial examples. We first formulate the kernel-prediction-based attack where an input image is convolved with kernels in a pixel-wise way, and the misclassification ability is guided by systematically tuning the kernel weight. In order to generate more natural motion blurred examples, we also propose the saliency-regularized adversarial kernel prediction, where the salient region serves as a moving object, and the predicted kernel is regularized to achieve visually natural effects. Besides, our method could easily adjust the blur effects of different exposure time during the image capturing in the real world, by adaptively tuning translations of object and background.

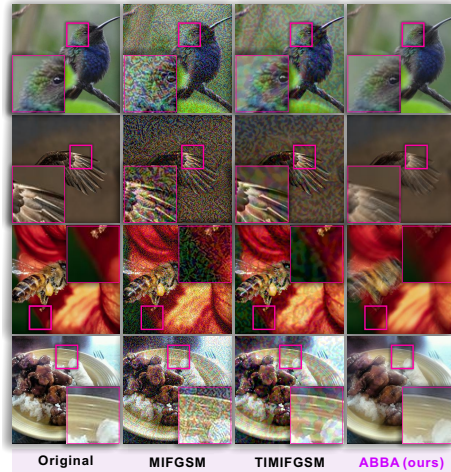

**Figure 1:** Four adversarial examples of MIFGSM [3], TIM-IFGSM [5] and our ABBA. MIFGSM and TIMIFGSM produces apparent noise on all four cases. Our ABBA generates visually natural motion blur. All adversarial images fool the Inception-v3 model.

We perform comprehensive evaluation on the effectiveness of the proposed ABBA, benchmarked against various noise-based attacks on both attack success rates and transferability. The main contributions of this work can be summarized as follows. ❶ To the best of our knowledge, we make the very first attempt to investigate kernel-based adversarial attack. ❷ We propose a motion-based adversarial blur attack as a new type of attack mode, to be added to the adversarial attack family. ❸ In order to produce more visually plausible blur attack, we introduce a saliency regularizer that forces consistent blur patterns within the boundary of the objects (or background in some cases). ❹ Compared with the state-of-the-art (SOTA) additive-noise based adversarial attacks and common blur techniques, our proposed method achieves better attack success rate and transferability. ❺ Furthermore, our proposed method has demonstrated higher penetration capability against the SOTA GAN-based deblur mechanism, compared to normal image motion blur.

**Related Work.** Since the discovery of adversarial examples to attack a deep neural network (DNN) both theoretically [1] and physically [2], there has been extensive research towards developing adversarial attack and defense mechanisms. The basic iterative method (BIM) [2], the C&W method [4], the fast gradient signed method (FGSM) [1], and the momentum iterative fast gradient sign method (MI-FGSM) [3], *etc*., are a few popular ones among early adopters in the research community. Building upon these ideas, researchers have been continuously pushing the envelope in many ways. For example, serious attempts have been made to integrate momentum term into the iterative process for the attacks [3]. By doing so, the momentum can help stabilize the update directions, begetting more transferable adversarial examples and posing more threats to adversarially trained defense mechanisms. More recently, [5] proposes to optimize the noise perturbation over an ensemble of translated images, making the generated adversarial examples more robust against white-box models being attacked while achieving better transferability. The mainstream adversarial attack is an additive

noise pattern that is learnable given the model parameters under a white-box setting. Perhaps the prevalence is partially due to the fact that the adversarial noise with the '*addition*' operation is relatively straightforward to optimize for. Of course, there are many other ways to alter a benign image beyond the addition operation that are all potential candidates for coming up with new types of adversarial attack modes. One caveat of additive noise attack is the lack of balance between being visually plausible and imperceptible while having high attack success rate. Usually, it has to compromise one for the other. Researchers are looking beyond additive noise attack to seek novel attack modes that strike a better balance between visual plausibility and performance, *e.g.*, coverage-guided fuzzing [6], multiplicative attack [7], deformation attack [8, 9, 10], and semantic manipulation attack [11].

We are proposing a new type of motion-based adversarial blur attack that can generate visually natural and plausible motion-blurred adversarial examples, inspired by kernel prediction [12, 13, 14] and motion blur generation [15]. One desired property of the proposed method is the immunity and robustness against the SOTA deblurring techniques (*e.g.*, [16, 17]). In an effort to understand black-box DNN better, an image saliency paradigm is proposed [18] to learn where an algorithm looks at, by discovering which parts of an image most affect output score when perturbed in terms of Gaussian blur, replacing the region with constant value and injecting noise. The localization of the blur region is performed through adaptive iteration whilst ours is saliency regularized that leads to a visually plausible motion-blurred image. The major difference is that their Gaussian blur kernel is fixed while ours is learnable to maximally jeopardize the image recognition DNNs.

## 2 Methodology

### 2.1 Background: Additive-Perturbation-Based Attack

Let $\mathbf{X}^{\text{real}}$ be a real (untampered) example, *e.g.*, images in the ImageNet dataset, and $y$ denotes its ground truth label. A classifier denoted as $f(\mathbf{X}) : \mathcal{X} \to \mathcal{Y}$ predicts the label of $\mathbf{X}^{\text{real}}$. An attack method aims to generate an adversarial example denoted as $\mathbf{X}^{\text{adv}}$ that can fool the classifier to predict an incorrect label with imperceptible perturbation. Existing attack methods mainly focus on the additive adversarial perturbation that is added to the real example to get $\mathbf{X}^{\text{adv}} = g(\mathbf{X}^{\text{real}}, \delta) = \mathbf{X}^{\text{real}} + \delta$, where $\delta$ is generated by maximizing a loss function $J(\mathbf{X}^{\text{adv}}, y)$ with a constrained term:

$$\arg\max_{\delta} J(\mathbf{X}^{\text{real}} + \delta, y) \text{ subject to } \|\delta\|_{\text{p}} \leq \epsilon_{\text{a}}, \tag{1}$$

For example, gradient descent is widely employed by many methods to generate adversarial examples, *e.g.*, FGSM, BIM, MIFGSM, DIM, and TIMIFGSM.

### 2.2 AB$\mathbb{B}$A$_{\text{pixel}}$: Kernel-Prediction-Based Adversarial Attack

Besides '+', there are various techniques that can perform advanced image transformation for different objectives, *e.g.*, Gaussian filter for image denoising, Laplacian filter for image sharpening, and guided filter for edge-preserving smoothing [19], which are all kernel-based techniques, processing each pixel of the image with a hand-crafted or guided kernel. In general, compared with the addition, kernel-based operation can handle more complex image processing tasks via different kinds of kernels. More recently, several works [20, 13, 14] have found that the kernel weights can be carefully predicted for advanced tasks with high performance, *e.g.*, high quality noise-free rendering and video frame interpolation. Inspired by these works, in this paper, we propose the kernel-prediction-based attack. Specifically, we process each pixel (*i.e.*, $\mathbf{X}_p^{\text{real}}$) of a real example $\mathbf{X}^{\text{real}}$ with a kernel $\mathbf{k}_p$,

$$\mathbf{X}_p^{\text{adv}} = g(\mathbf{X}_p^{\text{real}}, \mathbf{k}_p, \mathcal{N}(p)) = \sum_{q \in \mathcal{N}(p)} \mathbf{X}_q^{\text{real}} k_{pq}, \tag{2}$$

where $p$ denotes the $p$-th pixel in $\mathbf{X}^{\text{adv}}$ and $\mathbf{X}^{\text{real}}$, *i.e.*, $\mathbf{X}_p^{\text{adv}}$ and $\mathbf{X}_p^{\text{real}}$, respectively. $\mathcal{N}(p)$ is a set of $N$ pixels in $\mathbf{X}^{\text{real}}$ and $p \in \mathcal{N}(p)$. The kernel $\mathbf{k}_p$ also has the size of $N$ and determines the weights of $N$ pixels in $\mathcal{N}(p)$. In general, we have $\sum_{q \in \mathcal{N}(p)} k_{pq} = 1$ to ensure the generated image lies within the respective neighborhood of the input image, where a softmax activation is often adopted for this requirement [20]. To better understand Eq. (2), we discuss with two simple cases: 1) when we let $\mathcal{N}(p)$ be a neighborhood of the pixel $p$ and the kernel of each pixel is a fixed Gaussian kernel, $\mathbf{X}^{\text{adv}}$ is the Gaussian-blurred $\mathbf{X}^{\text{real}}$. Similarly, we can obtain defocus-blurred image with disk kernels. 2) when we set $\max(\mathbf{k}_p) = k_{pp}$ and $\forall q, k_{pq} \neq 0$, the perturbation of $\mathbf{X}^{\text{adv}}$ becomes more imperceptible as length of $\mathbf{k}_p$ decreases. To achieve high attack success rate, we need to optimize kernels of all

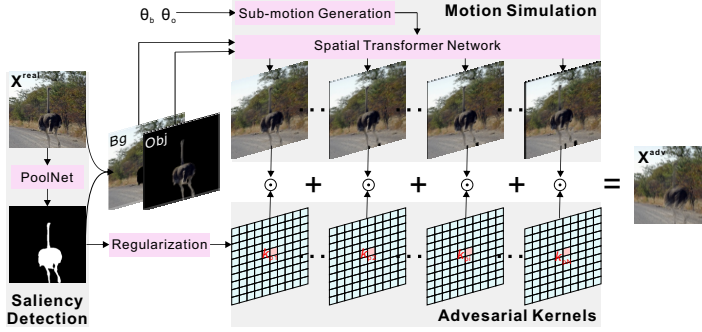

**Figure 3:** Pipeline of our motion-based adversarial blur attack, *i.e.*, Eq. (4). First, We use the PoolNet [21] to extract the salient object in the image and obtain the object and background regions. Then, the translation parameters $\theta_o$ and $\theta_b$ are divided to $N$ parts to simulate the motion process and generate $N$ images with the spatial transformer network [22]. Finally, we get the adversarial example by adding these images with adversarial kernels as weights for each pixel. The adversarial kernels and translation parameters are tuned to realize effective attack by optimizing Eq. (5). $\odot$ denotes the element-wise product.

pixels independently, *i.e.*, $\mathcal{K} = \{\mathbf{k}_p | \forall p \text{ in } \mathbf{X}^{\text{real}}\}$, according to the loss function of AI-related tasks, *e.g.*, image classification, and constrained terms

$$\arg\max_{\mathcal{K}} J(\{\sum_{q\in\mathcal{N}(p)} \mathbf{X}_q^{\text{real}} k_{pq}\}, y) \text{ subject to } \forall p, \|\mathbf{k}_p\|_0 \leq \epsilon, \max(\mathbf{k}_p) = k_{pp}, \sum_{q\in\mathcal{N}(p)} k_{pq} = 1, \quad (3)$$

where $\|\mathbf{k}_p\|_0$ represents the number of valid kernel elements (*i.e.*, $\{k_{pq} \neq 0\}$) and $\epsilon \in [1, N]$ controls the upper bound of $\|\mathbf{k}_p\|_0$. When $\epsilon = 1$, we have $\mathbf{X}^{\text{adv}} = \mathbf{X}^{\text{real}}$, and when $\epsilon = N$, the perturbation would be the most serious case. We can calculate the gradient of the loss function with respect to all kernels, to realize the gradient-based attack. As a result, the attack method can be integrated into any gradient-based additive-perturbation attack methods, *e.g.*, FGSM, BIM, MIFGSM.

Since the kernel-prediction-based adversarial attack tunes each pixel's kernel independently, it can achieve a significantly high attack success rate, but generating unnatural images that are easily perceptible. For an easier reference, we name this method ABBA$_{\text{pixel}}$. As shown in Fig. 2, ABBA$_{\text{pixel}}$ distorts the original inputs and produces additive noise-like results. To reach the balance between high attack success rate and natural

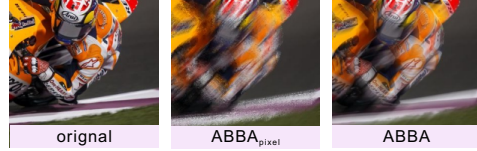

| orignal | ABBA$_{\text{pixel}}$ | ABBA |

**Figure 2:** From left to right: original image, adversarial examples generated by the kernel-prediction-based attack (ABBA$_{\text{pixel}}$), and motion-based adversarial blur attack (ABBA).

visual effect, we propose to regularize the kernels to produce visually natural motion blur via the guidance of a visual saliency map.

### 2.3 ABBA: Motion-Based Adversarial Blur Attack

#### 2.3.1 Saliency-Regularized Adversarial Kernel Prediction

Motion blur is often generated during the exposure time by integrating the light from a moving object. To synthesize the motion blur, we need to know where the object is and specify how it moves. To this end, given an image, we first use the SOTA saliency detection method, *i.e.*, PoolNet [21], to extract the salient object $\mathbf{S}$ from $\mathbf{X}^{\text{real}}$ and assume it is moving at the time when capturing the image. The saliency map $\mathbf{S}$ is a binary image, indicating the salient object region (*i.e.*, $\mathbf{X}^{\text{real}} \odot \mathbf{S}$) and background region (*i.e.*, $\mathbf{X}^{\text{real}} \odot (1 - \mathbf{S})$), as shown in Fig. 3. Then, we specify translation transformations to the object and background, respectively. We denote them as $\text{T}(\mathbf{X}^{\text{real}} \odot \mathbf{S}, \theta_o)$ and $\text{T}(\mathbf{X}^{\text{real}} \odot (1 - \mathbf{S}), \theta_b)$ that are simplified as $\mathbf{X}^{S,\theta_o}$ and $\mathbf{X}^{1-S,\theta_b}$, where $\theta_o$ and $\theta_b$ are the translation parameters[2] for the object and background, respectively.

Since motion blur is the integration of all light during the object moving process, we divide the motion represented by $\theta_o$ and $\theta_b$ into $N$ sub-motions (corresponding to the kernel size $N$ in Eq. (2)) to simulate blur generation. The sub-motions are represented by $\{i\Delta\theta_o, |i \in [1, N]\}$ and $\{i\Delta\theta_b, |i \in [1, N]\}$, where $\Delta\theta_o = \theta_o/N$ and $\Delta\theta_b = \theta_b/N$. Then, we redefine Eq. (2) as

$$\mathbf{X}_p^{\text{adv}} = \text{g}(\mathbf{X}_p^{\text{real}}, \mathbf{S}, \mathbf{k}_p, \mathcal{N}(p)) = \sum_{q=\mathcal{N}(p,i), i\in[1,N]} (\mathbf{X}_q^{S,i\Delta\theta_o} + \mathbf{X}_q^{1-S,i\Delta\theta_b}) k_{pq}, \quad (4)$$

where $\mathcal{N}(p)$ is a set of the $p$-th pixel in all translated examples. $\mathbf{X}^{S,i\Delta\theta_o}$ and $\mathbf{X}^{S,i\Delta\theta_o}$ denote the object and background images translated $i\Delta\theta_o$ pixels. Compared with the attack in Sec. 2.2, the perturbation amplitude is affected by the kernel and translation parameters. The objective function is defined as

$$\arg\max_{\mathcal{K},\theta_{\mathrm{o}},\theta_{\mathrm{b}}} J(\{ \sum_{\substack{q=\mathcal{N}(p,i)\\ i\in[1,N]}} (\mathbf{X}_q^{S,i\Delta\theta_{\mathrm{o}}} + \mathbf{X}_q^{1-S,i\Delta\theta_{\mathrm{b}}})k_{pq}\}, y) \quad (5)$$

$$\text{subject to } \forall p, \|\mathbf{k}_p\|_0 \leq \epsilon, \max(\mathbf{k}_p) = k_{pp}, \sum_{q\in\mathcal{N}(p)} k_{pq} = 1$$

$$\forall p, q, \mathbf{k}_p = \mathbf{k}_q, \text{ if } \mathbf{S}(p) = \mathbf{S}(q), \|\theta_{\mathrm{o}}\|_\infty \leq \epsilon_\theta, \|\theta_{\mathrm{b}}\|_\infty \leq \epsilon_\theta.$$

where $\epsilon_\theta \in [0,1]$ controls the maximum translations of the object/background. Here, we use the spatial transformer network [22] to perform translation according to $\theta_{\mathrm{o}}$ and $\theta_{\mathrm{b}}$, enabling the gradient propagate to all kernels. There are two main differences about the constrained terms *c.f.* Eq. (3): (1) The translation parameters are added to guide the generation of the adversarial example; (2) The kernels are set to be the same within the same region, which is needed to generate visually natural motion blur, since pixels in the object region usually have the same motion. As shown in Fig. 2, by incorporating saliency and motion regularization, the ABBA's adversarial example looks visually more natural than the one by ABBA$_{\mathrm{pixel}}$.

### 2.3.2 Attacking Algorithm

We summarize the workflow of our attacking algorithm in the following steps: 1) Calculate the saliency map of an image, *i.e.*, $\mathbf{X}^{\mathrm{real}}$, via PoolNet and obtain $\mathbf{S}$. 2) Initialize $\theta_{\mathrm{o},t} = \theta_{\mathrm{b},t} = [0,0]$ and set each kernel $\mathbf{k}_{p,t}$ of $\mathcal{K}_t$ by $\{k_{pp,t} = 1, k_{pq,t} = 0 | \forall q \in \mathcal{N}(p), q \neq p\}$ where $t = 0$ denotes the first iteration. 3) Calculate $\mathbf{X}_t^{\mathrm{adv}}$ via Eq. (4), which is also visualized in Fig. 3 for better understanding. 4) Calculate the gradient of $\mathbf{X}_t^{\mathrm{adv}}$ with respect to the objective function and obtain $\nabla_{\mathbf{X}_t^{\mathrm{adv}}} J(\mathbf{X}_t^{\mathrm{adv}}, y)$. 5) Propagate the gradient through the spatial transformer network and obtain the gradients of $\mathcal{K}_t$, $\theta_{\mathrm{o},t}$, and $\theta_{\mathrm{b},t}$, *i.e.*, $\nabla_{\mathcal{K}_t} J(\mathbf{X}_t^{\mathrm{adv}}, y)$, $\nabla_{\theta_{\mathrm{o},t}} J(\mathbf{X}_t^{\mathrm{adv}}, y)$ and $\nabla_{\theta_{\mathrm{b},t}} J(\mathbf{X}_t^{\mathrm{adv}}, y)$. 6) Update $\mathcal{K}_t$, $\theta_{\mathrm{o},t}$, and $\theta_{\mathrm{b},t}$ with a step size. 7) Update $t = t+1$ and go to the Step 3) for further optimization until it reaches the maximum iteration or $\mathbf{X}_t^{\mathrm{adv}}$ fools the DNN. We will detail our settings in Sec. 3.1.

### 2.4 ABBA$_{\mathrm{physical}}$: Towards Real-World Adversarial Blur Attack

As introduced in Sec. 2.3.2, ABBA takes a real example as the input and produces an adversarial blur example, the estimated kernels ($\mathcal{K}^*$), and translation parameters ($\theta_o^*$ and $\theta_b^*$). Then, it posts an interesting problem whether we could use the estimated translation parameters to guide camera or object moving, in generating a real-world adversarial blur examples. There are three main challenges: 1) We cannot control kernels' values in the real world. The optimized kernels are to let the adversarial examples fool deep models and may not exist in the real world. 2) It is difficult to precisely control the object or camera's moving without a high-precision robot arm. 3) To transfer the image translation to camera translation, we need know the object depth and camera intrinsic parameters.

To alleviate these challenges, we conduct the following modifications of our ABBA: 1) we fix the kernels $\mathcal{K}$ to be average kernels (*i.e.*, we set each kernel's elements as $\frac{1}{N}$ where $N$ is the kernel size) that let the generated adversarial blur follow the real-world motion blur[3]. 2) We force the object and background to share the translation parameters. As a result, ABBA simulates the blur generation through camera moving, where object and background have the same motion thus we can produce blurred examples by moving the camera. 3) In the real world, we could use a RGB-D camera to get an object's depth and leverage a calibration software to obtain the intrinsic parameters of the camera.

Based on these, we can generate a real-world adversarial blur example by: 1) capturing a picture containing an object in a real-world scene. 2) using our ABBA to calculate $\theta_o^*$ or $\theta_b^*$. 3) calculating the camera translations according to object depth and camera intrinsic parameters. 4) moving camera according to the camera translations and take a real blur picture during the moving process as the output. In particular, we will validate ABBA$_{\mathrm{physical}}$ in Sec. 3.4 through the AirSim simulator [25] within which we can control a simulated camera precisely and obtain the depth map. We also conduct an experiment with a mobile phone to preliminarily verify our method in the real world.

## 3 Experimental Results

### 3.1 Experimental Settings

**Dataset and Models.** We use NeurIPS'17 adversarial competition dataset [26], compatible with ImageNet, for all the experiments. To validate our method's transferability, we consider four widely used models, *i.e.*, Inception v3 (Inc-v3) [27], Inception v4 (Inc-v4), Inception ResNet v2 (IncRes-v2) [28], and Xception [29]. We further compare on four defense models: Inc-v3$_{\mathrm{ens3}}$, Inc-v3$_{\mathrm{ens4}}$, and IncRes-v2$_{\mathrm{ens}}$ from [30] and high-level representation guided denoiser (HGD) [31] with the highest ranking in NeurIPS'17 defense competition. We report the results of Inc-v3 in the paper, and put more results of other models (*e.g.*, Inc-v4, IncRes-v2, Xception) in the supplementary material.

| | Attacking Results (Inc-v3) | | | | Defence Results (Inc-v3) | | | |
|---|---|---|---|---|---|---|---|---|
| | Inc-v3 | Inc-v4 | IncRes-v2 | Xception | Inc-v3$_{env3}$ | Inc-v3$_{env4}$ | IncRes-v2$_{ens}$ | HGD |
| GaussBlur | 34.7 | 22.7 | 18.4 | 26.1 | 23.6 | 23.8 | 19.3 | 16.9 |
| GaussBlur$_{obj}$ | 13.6 | 6.0 | 5.2 | 7.1 | 8.6 | 7.8 | 6.3 | 4.6 |
| GaussBlur$_{bg}$ | 18.8 | 10.8 | 9.2 | 12.0 | 13.0 | 13.1 | 10.9 | 8.7 |
| DefocBlur | 30.0 | 16.8 | 11.1 | 18.8 | 17.5 | 18.3 | 15.0 | 12.9 |
| DefocBlur$_{obj}$ | 10.0 | 3.0 | 2.9 | 3.6 | 5.2 | 4.6 | 3.8 | 2.7 |
| DefocBlur$_{bg}$ | 16.9 | 9.2 | 7.0 | 10.5 | 10.1 | 10.3 | 9.2 | 7.8 |
| Interp$_{blur}$ | 34.7 | 3.6 | 0.5 | 3.4 | 7.1 | 7.1 | 4.3 | 1.4 |
| ABBA$_{obj}$ | 21.0 | 4.9 | 4.2 | 7.0 | 10.1 | 10.5 | 8.3 | 4.9 |
| ABBA$_{bg}$ | 30.9 | 11.6 | 10.1 | 12.9 | 1.2 | 0.8 | 1.2 | 0.5 |
| ABBA$_{image}$ | 62.4 | 29.8 | 28.8 | 34.1 | 43.2 | 43.8 | 38.9 | 28.4 |
| ABBA$_{pixel}$ | 89.2 | 65.5 | 65.8 | 71.2 | 69.8 | 72.5 | 68.0 | 63.1 |
| ABBA | 65.6 | 31.2 | 29.7 | 33.5 | 46.6 | 48.7 | 41.2 | 31.0 |
| Interp$_{noise}$ | 95.8 | 20.5 | 15.6 | 22.9 | 16.8 | 16.1 | 9.4 | 3.3 |
| FGSM | 79.6 | 35.9 | 30.6 | 42.1 | 15.6 | 14.7 | 7.0 | 2.1 |
| MIFGSM | 97.8 | 47.1 | 46.4 | 47.7 | 20.5 | 17.4 | 9.5 | 6.9 |
| DIM | 98.3 | 73.8 | 67.8 | 71.6 | 24.2 | 24.3 | 13.0 | 9.7 |
| TIFGSM | 75.4 | 37.3 | 32.1 | 38.6 | 28.2 | 28.9 | 22.3 | 18.4 |
| TIMIFGSM | 97.9 | 52.4 | 47.9 | 44.6 | 35.8 | 35.1 | 25.8 | 25.7 |
| TIDIM | 98.5 | 75.2 | 69.2 | 61.3 | 46.9 | 47.1 | 37.4 | 38.3 |

**Table 1:** Adversarial comparison results on NeurIPS'17 adversarial competition dataset according to the success rate. The adversarial examples are generated from Inc-v3. There are two comparison groups. For the first one, we compare blur-based methods, *i.e.*, Interpretation-based blur (Interp$_{blur}$), GaussBlur, and DefocusBlur with our **ABBA** by considering the effects of attacking different regions, *i.e.*, object or background regions, of inputs. In addition to above methods, the second group comparison contains additive-perturbation-based attacks, *i.e.*, Interpretation-based noise (Interp$_{noise}$) [18], FGSM [1], MIFGSM [3], DIM [32], and TIFGSM, TIM-IFGSM, and TIDIM [5]. We highlight the top three results with pink, yellow, and blue, respectively.

**Baselines.** We consider two kinds of baselines. The first scope is SOTA additive-perturbation-based attacks, *e.g.*, FGSM [1], MIFGSM [3], DIM [32], TIFGSM, TIMIFGSM, and TIDIM [5][4], and interpretation-based noise [18]. The second kind contains three blur-based methods including the interpretation-based blur [4], Gaussian blur [33], and Defocus blur. For the transferability comparison in Sec. 3.2, we follow the default settings in [5] for the first group attacks. For all iterative attack methods including ours, we set the iteration number to be 10. For blur-based baselines, we set the standard variation of Gaussian blur and the kernel size of Defocus blur to be 15.0, which is the same with our method for a fair comparison. For the image quality comparison in Sec. 3.2, we tune the hyper-parameter of all attacks to cover the success rate from low to high and show the relationship between image quality and the success rate. This helps to see if our method could maintain high attack success rate and transferability while keeping the image natural.

**Setup of ABBA.** In the experimental part of Sec. 3.2, we implement our methods by setting the hyper-parameters, *i.e.*, maximum translation $\epsilon_\theta$ to 0.4 and maximum valid kernel size $\epsilon$ to 15.0 with the iteration 10. The step sizes are set to $0.04$ and $1.5$ for updating the kernels and translation parameters, respectively. Such setups are around the medium values among the range of our hyper-parameters, well balancing the attack success rate and visual quality. We discuss the effect of $\epsilon_\theta$ and $\epsilon$ in Sec. 3.6. In addition, we have four variants of our method, *i.e.*, ABBA$_{pixel}$, ABBA$_{obj}$, ABBA$_{bg}$, and ABBA$_{image}$. The first one is the attack introduced in Sec. 3.6 while the rest three methods blur different regions of input images, *e.g.*, ABBA$_{obj}$ only adds motion blur to the object region by fixing the kernels of background pixels to be $\{k_{pp} = 1, k_{pq} = 0 | \mathbf{S}(p) = 0, q \in \mathcal{N}(p), q \neq p\}$ and ABBA$_{image}$ adds motion blur to the whole image while forcing object and background to share the kernels and translations.

**Metrics.** We use the success rate (Succ. Rate), *i.e.*, the rate of adversarial examples that fool attacked DNNs, to evaluate the effectiveness of the attacks. Regarding quality of generated adversarial examples, we use BRISQUE [34] instead of $L_p$ norms, PSNR, or SSIM due to the following reasons: 1) our ABBA cannot be fairly evaluated by the $L_1$, $L_2$, $L_\infty$, PSNR, or SSIM metrics since the perturbation is not additive and no longer well aligned pixel-to-pixel. 2) BRISQUE [34] is a natural scene statistic-based distortion-generic blind/no-reference (NR) image quality assessment and widely used to evaluate losses of 'naturalness' in the image due to the presence of distortions including additive noise and blur. Let it be noted that a more natural image often has a smaller BRISQUE value.

## 3.2 Comparison with Baselines on Transferability

Tab. 1 summarizes the comparison results and we discuss from two aspects: 1) the comparison with additive-perturbation-based attacks. 2) the advantages over blur-based methods. For the first aspect, compared with most of the additive-perturbation-based attacks, our method, *i.e.*, ABBA and ABBA$_{pixel}$, achieve much higher success rate on all defences models, demonstrating the higher transferability of ABBA over baselines. Compared with two SOTA methods, *i.e.*, DIM and TIDIM, ABBA$_{pixel}$ achieves slightly lower success rate when attacking Inc-v4 and IncRes-v2 while obtaining higher results than TIDIM when attacking Xception. In summary, our methods ABBA$_{pixel}$ and ABBA have competitive transferability with SOTA additive-perturbation-based attacks while achieving significant advantages in attacking defence models. For the second aspect, ABBA achieves higher success rate than GaussBlur, DefocusBlur, and Interp$_{blur}$ on all normally trained and defense models. We also compare them for attacking object or background regions. Obviously, the success rates of all methods decrease significantly when we add blur to only the object or background regions. Besides these transferability results, we also add an analysis in the supplementary material about the interpretable explanation of the high transferability of our method by comparing with FGSM and MIFGSM.

## 3.3 Comparison with Baselines on Image Quality

We conduct an analysis about the attack success rate and image quality (*i.e.*, measured by BRISQUE [34] that evaluates losses of 'naturalness' in the image). Note that, smaller BRISQUE corresponds to more natural images. We also consider two comparison groups: 1) the blur-based attacks, *i.e.*, GaussBlur and DefocBlur, and 2) SOTA additive-perturbation-based attacks, *i.e.*, FGSM, MIFGSM, DIM, TIFGSM, TIMIFGSM, and TIDIM. For each compared method, we tune their hyper-parameters to cover success rates from low to high on the NeurIPS'17 adversarial competition dataset. For each hyper-parameter, we calculate the average BRISQUE of the adversarial images that successfully fool DNNs. As a result, for each attacked model, we can draw a plot for an attack method and the points more near the top left corner are better (*i.e.*, looks more naturally while fooling DNNs).

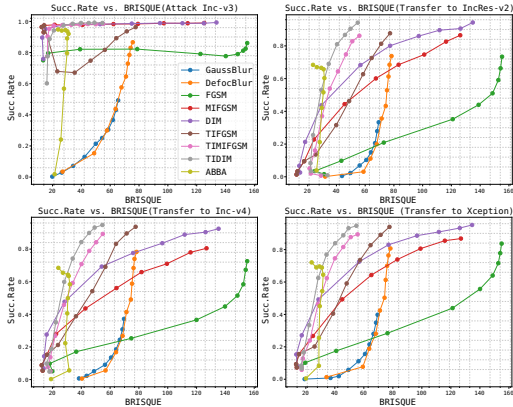

As shown in Fig. 4, in general, the quality of images generated by all baseline methods gradually gets worse as their success rate becomes larger. In contrast, the BRISQUE of our ABBA always stay at a small value even if the success rate increases, demonstrating that ABBA can produce visually natural adversarial examples with high attack success rate. When transferring the adversarial examples to other models, we observe that the success rate of additive-perturbation-based attacks, especially the FGSM, MIFGSM, TIFGSM, and DIM, decrease sharply, while the blur-based attacks are not impacted so largely. Comparing with SOTA methods (*i.e.*, TIDIM and TIMIFGSM) on the transferability, we see that the adversarial examples of ABBA and the two methods have similar

**Figure 4:** Succ. Rate vs. BRISQUE. Note that, smaller BRISQUE corresponds to more natural images.

BRISQUE scores when their success rate is small. When success rate of TIDIM and TIFGSM further increases, their BRISQUE values become smaller than ours.

## 3.4 Adversarial Blur Examples in the Simulation World and Real World

**Results in AirSim environment:** We validate our ABBA_physical on the AirSim simulator [35] that supports hardware-in-loop with cameras for physically and visually realistic simulations. AirSim also provides APIs to retrieve relevant data (*e.g.*, real-time depth map) and controls cameras in a platform independent way, which meets the requirements of ABBA_physical introduced in Sec. 2.4. In particular, we choose the open-source Neighborhood environment [36], containing 70 cars with various styles. For each car, we select a observation view where a DNN (*e.g.*, Inc-v3) could classify it correctly. Then, we conduct the ABBA_physical experiment for each car with the following steps: 1) Set a camera to capture an original image of the car at the selected observation view. 2) Use our method in-

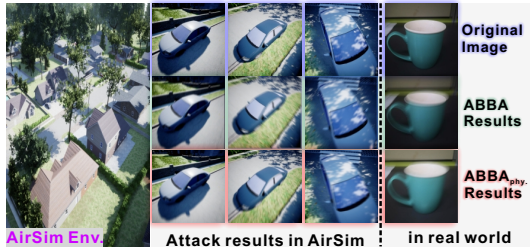

**Figure 5:** Attack results in the AirSim environment (medium) and real world (right). The left sub-figure shows a snapshot of the neighborhood environment. The results of ABBA_physical are produced by physically moving the camera and mobile phone in the AirSim and real world, respectively. All the adversarial examples fool the Inc-v3 model.

troduced in Sec. 2.4 to calculate an adversarial blurred image (*i.e.*, ABBA's result) and the camera translation with the depth map and camera intrinsic parameters. 3) Move the camera along a straight line to the translation destination while taking $N$ pictures and averaging them to the final blurred image (*i.e.*, ABBA_physical's result), which is equivalent to the motion blur generation process [23]. 4) Test whether the real blurred image and adversarial blurred image could fool DNNs (*i.e.*, Inc-v3, Inc-v4, IncRes-v2, and Xception in Table 2) successfully. We summarize the attack success rates in Table 2 and observe that: 1) ABBA_physical generated adversarial blur examples by physically moving cameras can achieve high success rate on Inc-v3, which demonstrates that there might exist real-world motion blur that can fool DNNs easily. 2) Although our ABBA has very high transferability across DNNs, the physical adversarial examples cannot fool other DNNs easily. **Results in the real world environment:** We further perform a preliminary experiment to validate our ABBA_physical in the real world through a mobile phone: 1) we capture a sharp image with a mobile phone, *i.e.*, the cup in the first row of Fig. 5. 2) we use ABBA to generate an adversarial blur image with the Inc-v3 model, *i.e.*, the second row of Fig. 5, and obtain the image translation parameters that indicate the mobile phone's moving direction

and distance. 3) we move the mobile phone along the direction indicated by the translation parameters and shoot a real-blurred image in the same scene with longer exposure time. We find both blurred images are misclassified by the Inc-v3 model. Here, we ignore the cup's depth since we shoot the sharp and real-blurred images at almost the same position and empirically tune the moving distance of the mobile phone. Such operation could be replaced by high-precision robot arms in the future.

### 3.5 Effect of Deblurring Methods

Here, we discuss the effect of SOTA deblurring methods to our adversarial examples and 'normal' motion blurred images. The 'normal' motion blur is usually synthesized by averaging neighbouring video frames [23], which is equivalent to setting the kernel weights as $\frac{1}{N}$ to average the translated examples in Eq. (4). We can regard such

**Table 2:** Success rate of ABBA and ABBA$_{physical}$ for attacking DNNs in the AirSim environment.

| | Adversarial examples from Inc-v3 | | | |
|---|---|---|---|---|
| | Inc-v3 | Inc-v4 | IncRes-v2 | Xception |
| Succ. Rate of ABBA | 97.6 | 87.8 | 82.9 | 90.2 |
| Succ. Rate of ABBA$_{physical}$ | 85.3 | 9.7 | 14.6 | 11.0 |

normal motion blur as an attack and add them to all images in the testing data. We use DeblurGAN [16] and DeblurGANv2 [17] to handle our adversarial examples and the normal motion blurred images and calculate the relative decrease of the success rate, *i.e.*, $r = \frac{s-s'}{s}$ where $s$ and $s'$ represent the success rate before and after deblurring. Smaller $r$ means that the attack is more resilient against deblurring methods. As shown in Fig. 6, in general, compared with the normal motion blurred images, our adversarial examples are harder to be handled by the state-of-the-art deblurring methods. For DeblurGANv2, the relative decrease $r$ of normal motion blur is usually larger than 0.5 when kernel size is in [15, 35]. This means that DeblurGANv2 can effectively remove the normal motion blur and improve the classification accuracy. In contrast, the $r$ of our method is always smaller than the $r$ of normal motion blur, and gradually decreases in the range of [15, 35], indicating that it is more difficult to use DeblurGANv2 to defend our attack as kernel size becomes larger. Similar results are obtained on DeblurGAN where the difference of $r$ between our method and normal motion blur is much smaller than that on DeblurGANv2.

### 3.6 Hyper-parameter Analysis and Ablation Study

**Effect of $\epsilon$ and $\epsilon_\theta$:** We calculate the success rate of our method with different $\epsilon$ and $\epsilon_\theta$ in Eq. (5) and find that the success rates become gradually higher with the increase of $\epsilon$ and $\epsilon_\theta$. More results and discussions can be found in the supplementary material. **Effect of motion directions:** we fix $\epsilon_\theta$ and $\epsilon$ and tune the motion direction of object and background by setting different x-axis and y-axis translations to see the success rate variation. We find that the success rate reaches the highest value around $45°$ moving direction. We visualize and discuss the results in the supplementary material.

**Effect of blurred regions and importance of adaptive translations:** We conduct an ablation study by adding motion blur to different regions of an image. The success rate results are reported in Tab. 1. Compared with ABBA$_{pixel}$, ABBA strikes a good balance between the attack success rate and visual effects. Compared with other variants, ABBA that jointly tunes the object and background translations can obtain much better transferability across normally trained and defense-based models. We show and discuss their visualization results in the supplementary material.

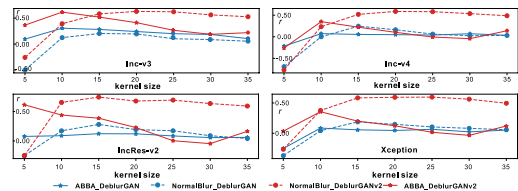

**Figure 6:** The relative decrease of attack success rate before and after deblurring. Two state-of-the-art deblurring methods, *i.e.*, DeblurGAN [16] and DeblurGANv2 [17], are used to deblur the our adversarial blurred examples and normal motion blurred images.

## 4 Conclusions

In this paper, we have initiated the first step to comprehensively investigate the potential hazards of motion blur for DNNs. We propose the kernel-prediction-based attack that can fool DNNs with a high success rate, which is further regularized by visual saliency to make the motion-blurred image visually more natural. Besides, we also validate that our adversarial blur might indeed exist in the real world. Comprehensive evaluation demonstrates the usefulness of our methods. Our results call for the attention of future research that takes motion blur effects into consideration during real-time image perception DNN designs. We also hope that our work facilities more general solutions of robust DNN towards addressing common camera motion blur effects. Moreover, we have demonstrated that the proposed kernel-prediction-based adversarial attack can be extended to other kinds of attacking methods, *e.g.*, adversarial attacks based on denoising [37], raining [38], camera exposure [39], and bias field in the medical imaging [40]. We will also discuss the effects of motion blur to visual object tracking [41, 42, 43], through the proposed ABBA and recent attacking method [44] against tracking.

## 5   Broader Impact

In this work, we make an early attempt to investigate the motion-blur effects to DNNs, which is a common phenomenon in the real-world image capturing process of a camera. We present the very *first* attack based on manipulating the motion blur of the images. Through comprehensive experiments, we have demonstrated that very successful attacks can be well disguised in naturally-looking motion blur patterns of the image, unveiling the system vulnerabilities of any image capture stage that deals with camera or object motions.

Considering that image capturing and sensing is an integral and essential part of almost every computer vision application that interacts with the real world, the message we are trying to convey here is an important one, *i.e.*, attackers can intentionally make use of motion blur, either by tampering with the camera sensing hardware or the image processing software to embed such an attack. Even unintentionally, the motion blur effects still commonly exist in the real-world application, posting threats to the DNNs behind the camera. This work is the first attempt to identify and showcase that such an attack based off image motion blur is not only feasible, but also leads to high attack success rate while simultaneously maintaining high realisticity in the image motion blur patterns. In a larger sense, this work can and will provide new thinking into how to better design the image capturing pipeline in order to mitigate potential risk caused by the vulnerabilities discussed herein, especially for mission- and safety-critical applications that are involved with moving objects or moving sensors such as autonomous driving scenarios, mobile face authentication with a hand-held device, computer-aided diagnostics in medical imaging, robotics, *etc*.

Bad actors can potentially make use of this newly proposed attack mode as a wheel to pose risks on existing imaging systems that are not yet prepared for this new type of attack and effect based on image motion blur. We, as researchers, believe that our proposed method can accelerate the research and development of the DNN resilient mechanism against such motion blur effects. Therefore, our work can serve as an asset and a stepping stone for future-generation trustworthy design of computer vision DNNs and systems.

In addition to the societal impact discussed above, the proposed method can also influence various research directions. For example, our proposed ABBA method:

- hints new data augmentation technique for training powerful DNN-based deblurring methods.
- hints new DNN design, detection/defense techniques to be resilience against motion blur effects.
- hints new direction of analyzing the effect of motion blur to video analysis tasks, *e.g.*, real-time visual object detection, tracking, segmentation, and action recognition.

## Acknowledgments and Disclosure of Funding

We appreciate the anonymous reviewers for their valuable feedback. This research was supported in part by Singapore National Cy-bersecurity R&D Program No. NRF2018NCR-NCR005-0001, Na-tional Satellite of Excellence in Trustworthy Software System No.NRF2018NCR-NSOE003-0001, NRF Investigatorship No. NRFI06-2020-0022. It was also supported by JSPS KAKENHI Grant No.20H04168, 19K24348, 19H04086, and JST-Mirai Program Grant No.JPMJMI18BB, Japan, and the National Natural Science Foundation of China (NSFC) under Grant 61671325 and Grant 62072334. We gratefully acknowledge the support of NVIDIA AI Tech Center (NVAITC) to our research.

## Footnotes

*Xiaofei Xie and Lei Ma are corresponding authors (xfxie@ntu.edu.sg, malei@ait.kyushu-u.ac.jp).

[2]$\theta_o$ and $\theta_b$ are 2D vectors in the range of [0,1] and represent the rates of x-axis and y-axis shifting distance of the object&background regions w.r.t. image size.

[3]We use the average kernels to simulate the real motion blur generation process, which has been used to construct training dataset for deblurring methods [23, 24].

[4] We use the released code in github.com/dongyp13/Translation-Invariant-Attacks to get the results of FGSM, MIFGSM, DIM, TIFGSM, TIMIFGSM, and TIDIM.

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
