[Supplementary Material]

# Watch out! Motion is Blurring the Vision of Your Deep Neural Networks

## *Supplementary Material*

**Qing Guo**[1]     **Felix Juefei-Xu**[2]     **Xiaofei Xie**[1]     **Lei Ma**[3]

**Jian Wang**[1]     **Bing Yu**[3]     **Wei Feng**[4]     **Yang Liu**[1]

[1]Nanyang Technological University, Singapore     [2]Alibaba Group, USA
[3]Kyushu University, Japan     [4]Tianjin University, China

In the main paper [1] [1], we have reported the attack results of Inc-v3 on four normal trained models and four defense models, and compared with 14 attack instances on the transferability and the image quality. In this supplementary material, (1) we present the evaluation details of our method regarding the transferability on five more defense models, comparing the visualization results with SOTA attacks, and discussing the attack results of another three DNNs. (2) We also conducted an in-depth hyper-parameter analysis and ablation study of our method, and posted an interpretable explanation about the difference between our method and baselines on the transferability. (3) We validated the generalization of our method by attacking an STN-based CNN. (4) We demonstrated that our method could help enhance the blur robustness of DNNs for the classification task with the results on ImageNetC. (5) We show more adversarial attack results in the real world. (6) We discuss the defense results via re-trained DeblurGANv2 with the blurred images from our methods.

Overall, the results of this supplementary material further demonstrated that the proposed adversarial blur attack can fool DNNs effectively while generating visually natural blurred images. All experimental results and discussions infer that motion blur as a common effect in the real world has a high risk of fooling SOTA DNNs and our attack methods initiate the first step to study the potential hazards of motion blur for DNNs.

## 1 Attack Results on Eight Defense Models

Besides the results on the four defense models reported in the main paper, we also compared our method with baselines on another five defense models including R&P [2], NeurIPS-r3[2], and three models from the stae-of-the-art feature denoise-based (FD) defense method [3] (*i.e.*, ResNetXt101 with all denoising ($FD_{R101}$), ResNet152 with four denoising blocks ($FD_{R152}$), and adversarial trained baseline model ResNet152 ($FD_{R152B}$)). The R&P method transforms input images through random resizing and padding, which ranked the second in the NeurIPS 2017 defense competition. NeurIPS-r3 is the third rank submission of NeurIPS 2017 defense competition and combines adversarial trained VGG16, Inc-v3, IncRes-v2, and ResNet152v2 models in an ensemble way. Besides, NeurIPS-r3 also performs transformations, *i.e.*, shear, shifting, zoom, rotation, JPEG compression, and noise corruption, on input images. The FD method ranked the first in Competition on Adversarial Attacks and Defenses (CAAD)-2018. Note that, all other results of above baselines in our main paper and supplementary material are based on $L_\infty$ norm bound.

**Figure I.** Three visualization results of ABBA_pixel, ABBA, DIM, and TIDIM. All adversarial examples mislead the Inc-v3 model.

As reported in Table I, our method, *i.e.*, ABBA_pixel, achieves the highest transferability across all defense models and ABBA has competitive results with the state-of-the-art baseline TIDIM. Such results reveal a potential big shortcoming of existing studies of defense methods, *i.e.*, only considering the adversarial noise while ignoring other potential factors in physical environment. Note that, compared with adversarial noise, motion blur frequently happens in our daily life and widely exists among various computer vision-based applications, thus its influence to DNNs should be carefully studied and addressed.

**Table I.** Adversarial comparison results on NeurIPS'17 adversarial competition dataset according to the success rate. We use nine defense models to evaluate all attacks. The adversarial examples are generated from Inc-v3. There are two comparison groups. For the first one, we compare blur-based methods, *i.e.*, Interpretation-based blur (Interp_blur), GaussBlur, and DefocusBlur with our **ABBA** by considering the effects of attacking different regions, *i.e.*, object or background regions, of inputs. In addition to above methods, the second group comparison contains additive-perturbation-based attacks, *i.e.*, Interpretation-based noise (Interp_noise) [4], FGSM [5], MIFGSM [6], DIM [7], and TIFGSM, TIMIFGSM, and TIDIM [8]. We highlight the top three results with pink , yellow , and blue , respectively.

| | Defence Results (Adv. Examples from Inc-v3) | | | | | | | | |
| | Inc-v3_env3 | Inc-v3_env4 | IncRes-v2_ens | HGD | R&P | NeurPIS-r3 | FD_R101 | FD_R152 | FD_R152B |
|---|---|---|---|---|---|---|---|---|---|
| GaussBlur | 23.6 | 23.8 | 19.3 | 16.9 | 17.2 | 17.6 | 35.6 | 35.8 | 35.9 |
| GaussBlur_obj | 8.6 | 7.8 | 6.3 | 4.6 | 4.8 | 5.1 | 13.9 | 13.9 | 14.6 |
| GaussBlur_bg | 13.0 | 13.1 | 10.9 | 8.7 | 10.0 | 9.3 | 19.5 | 19.2 | 20.1 |
| DefocBlur | 17.5 | 18.3 | 15.0 | 12.9 | 14.6 | 14.2 | 31.1 | 30.9 | 31.1 |
| DefocBlur_obj | 5.2 | 4.6 | 3.8 | 2.7 | 3.3 | 2.9 | 10.8 | 10.5 | 11.1 |
| DefocBlur_bg | 10.1 | 10.3 | 9.2 | 7.8 | 9.0 | 8.1 | 19.5 | 17.6 | 18.5 |
| Interp_blur | 7.1 | 7.1 | 4.3 | 1.4 | 2.9 | 2.9 | 25.5 | 25.8 | 28.6 |
| ABBA_obj | 10.1 | 10.5 | 8.3 | 4.9 | 6.2 | 7.1 | 18.7 | 18.4 | 19.1 |
| ABBA_bg | 1.2 | 0.8 | 1.2 | 0.5 | 0.6 | 0.7 | 43.5 | 44.1 | 45.5 |
| ABBA_image | 43.2 | 43.8 | 38.9 | 28.4 | 34.1 | 35.0 | 61.1 | 61.9 | 62.4 |
| ABBA_pixel | 69.8 | 72.5 | 68.0 | 63.1 | 65.0 | 65.7 | 79.6 | 81.0 | 82.1 |
| ABBA | 46.6 | 48.7 | 41.2 | 31.0 | 36.7 | 38.5 | 64.2 | 64.6 | 65.6 |
| Interp_noise | 16.8 | 16.1 | 9.4 | 3.3 | 4.1 | 4.4 | 39.6 | 41.4 | 46.8 |
| FGSM | 15.6 | 14.7 | 7.0 | 2.1 | 6.5 | 9.8 | 39.2 | 41.4 | 45.3 |
| MIFGSM | 20.5 | 17.4 | 9.5 | 6.9 | 8.7 | 12.9 | 39.0 | 40.2 | 44.6 |
| DIM | 24.2 | 24.3 | 13.0 | 9.7 | 13.3 | 18.0 | 39.1 | 40.3 | 45.1 |
| TIFGSM | 28.2 | 28.9 | 22.3 | 18.4 | 19.8 | 24.5 | 39.7 | 41.8 | 45.4 |
| TIMIFGSM | 35.8 | 35.1 | 25.8 | 25.7 | 23.9 | 26.7 | 39.3 | 41.2 | 45.8 |
| TIDIM | 46.9 | 47.1 | 37.4 | 38.3 | 36.8 | 41.4 | 40.0 | 42.2 | 45.8 |

## 2 Visualization Comparison with Baselines

We show several adversarial examples of ABBA_pixel, ABBA, DIM, and TIDIM in Fig. I, Fig. IX and Fig. X. All examples can mislead the Inc-v3 model.

Obviously, our method ABBA can generate visually natural motion-blurred examples on various objects and these examples are very similar to real images captured by real-world cameras where the motion blur is caused by object or camera moving. In contrast, the adversarial examples of DIM and TIDIM have obvious unreal patterns. The noise-like pattern of DIM is drastically different from the natural noise usually caused by the camera sensor, *e.g.*, Gaussian noise. The perturbation pattern of TIDIM is more perceptible than that of DIM, although TIDIM achieves much higher transferability than DIM. Compared with ABBA, our another method, *i.e.*, ABBA_pixel, breaks local pattern of the original input. However, ABBA_pixel's examples look more imperceptible than TIDIM's results. More comparison results are shown in Fig. IX and Fig. X.

**Figure II.** Comparison between adversarial-blurred images and blurred images for training deblurring models.

**Table II.** Adversarial comparison results on NeurIPS'17 adversarial competition dataset. There is no available Xception model based on the author's implementations [8] of baselines, *i.e.*, FGSM, MIFGSM, DIM, TIFGSM, TIFMIFGSM, and TIDIM. Hence, we leave these baselines' results empty for the Xception model. We highlight the top three results with pink , yellow , and blue , respectively.

| | Attacking Results (Inc-v3) | | | | Attacking Results (Inc-v4) | | | | Attacking Results (IncRes-v2) | | | | Attacking Results (Xception) | | | |
|---|---|---|---|---|---|---|---|---|---|---|---|---|---|---|---|---|
| | Inc-v3 | Inc-v4 | IncRes-v2 | Xception | Inc-v3 | Inc-v4 | IncRes-v2 | Xception | Inc-v3 | Inc-v4 | IncRes-v2 | Xception | Inc-v3 | Inc-v4 | IncRes-v2 | Xception |
| GaussBlur | 34.7 | 22.7 | 18.4 | 26.1 | 14.2 | 26.7 | 10.9 | 17.2 | 12.1 | 11.8 | 20.1 | 13.8 | 16.1 | 15.7 | 11.9 | 32.5 |
| GaussBlur$_{obj}$ | 13.6 | 6.0 | 5.2 | 7.1 | 3.5 | 9.5 | 2.2 | 3.9 | 3.2 | 2.8 | 6.4 | 2.7 | 3.7 | 3.4 | 2.6 | 10.9 |
| GaussBlur$_{bg}$ | 18.8 | 10.8 | 9.2 | 12.0 | 6.7 | 13.4 | 5.5 | 7.3 | 6.7 | 6.5 | 11.8 | 6.8 | 7.6 | 7.1 | 6.3 | 16.3 |
| DefocBlur | 30.0 | 16.8 | 11.1 | 18.8 | 18.7 | 36.2 | 13.2 | 22.3 | 15.8 | 14.7 | 23.4 | 17.4 | 18.5 | 18.7 | 12.9 | 36.8 |
| DefocBlur$_{obj}$ | 10.0 | 3.0 | 2.9 | 3.6 | 3.9 | 10.3 | 3.1 | 4.4 | 3.8 | 3.2 | 7.5 | 3.2 | 4.4 | 4.4 | 2.6 | 11.8 |
| DefocBlur$_{bg}$ | 16.9 | 9.2 | 7.0 | 10.5 | 10.4 | 20.1 | 8.3 | 12.8 | 8.9 | 9.4 | 15.3 | 10.1 | 10.2 | 11.4 | 8.4 | 21.6 |
| Interp$_{blur}$ | 34.7 | 3.6 | 0.5 | 3.4 | 2.7 | 26.7 | 0.8 | 3.1 | 3.1 | 3.1 | 20.1 | 3.4 | 3.0 | 3.1 | 0.8 | 32.5 |
| ABBA$_{obj}$ | 21.0 | 4.9 | 4.2 | 7.0 | 11.6 | 28.9 | 9.7 | 11.5 | 11.2 | 11.9 | 29.0 | 12.7 | 9.1 | 9.6 | 7.7 | 30.2 |
| ABBA$_{bg}$ | 30.9 | 11.6 | 10.1 | 12.9 | 14.0 | 31.7 | 13.3 | 15.7 | 14.0 | 14.0 | 25.8 | 13.2 | 12.5 | 14.3 | 11.4 | 33.3 |
| ABBA$_{image}$ | 62.4 | 29.8 | 28.8 | 34.1 | 32.0 | 66.7 | 28.8 | 36.2 | 33.0 | 30.7 | 63.4 | 37.0 | 28.9 | 28.4 | 26.1 | 66.7 |
| ABBA$_{pixel}$ | 89.2 | 65.5 | 65.8 | 71.2 | 77.7 | 88.1 | 71.3 | 76.0 | 81.8 | 78.3 | 92.0 | 80.6 | 74.0 | 67.5 | 66.8 | 86.2 |
| ABBA | 65.6 | 31.2 | 29.7 | 33.5 | 39.5 | 74.9 | 37.3 | 43.2 | 38.4 | 38.6 | 71.6 | 44.2 | 32.3 | 35.2 | 35.9 | 73.1 |
| Interp$_{noise}$ | 95.8 | 20.5 | 15.6 | 22.9 | 5.2 | 92.6 | 1.6 | 6.0 | 6.7 | 6.0 | 91.8 | 8.3 | 3.5 | 2.3 | 0.4 | 93.4 |
| FGSM | 79.6 | 35.9 | 30.6 | 42.1 | 43.1 | 72.6 | 32.5 | 45.2 | 44.3 | 36.1 | 64.3 | 45.4 | | | | |
| MIFGSM | 97.8 | 47.1 | 46.4 | 47.7 | 67.1 | 98.8 | 54.3 | 58.5 | 74.8 | 64.8 | 100.0 | 61.7 | | | | |
| DIM | 98.3 | 73.8 | 67.8 | 71.6 | 81.8 | 98.2 | 74.2 | 79.1 | 86.1 | 83.5 | 99.1 | 80.8 | | | | |
| TIFGSM | 75.4 | 37.3 | 32.1 | 38.6 | 45.3 | 68.1 | 33.7 | 39.4 | 49.7 | 41.5 | 63.7 | 44.0 | | | | |
| TIMIFGSM | 97.9 | 52.4 | 47.9 | 44.6 | 68.6 | 98.8 | 55.3 | 50.8 | 76.1 | 69.5 | 100.0 | 59.9 | | | | |
| TIDIM | 98.5 | 75.2 | 69.2 | 61.3 | 80.7 | 98.7 | 73.2 | 65.5 | 86.4 | 85.5 | 98.8 | 71.0 | | | | |

Besides above visualization results, we further conduct an experiment to compare our adversarial blur images with the blur images for training deblurring models [9]. Specifically, given a sharp image, *e.g.*, the left sub-figures in Fig. II, we use ABBA to generate corresponding adversarial blur images and compare them with the blur images for training. Obviously, both blur looks realistic, which demonstrates the capability of ABBA to generate visually natural blur images.

# 3 Attack Results of Inc-v3, Inc-v4, IncRes-v2, and Xception

Besides the attack results of Inc-v3 reported in our main paper, we further show the results of Inc-v4, IncRes-v2, and Xception in Table II. Note that, there is no available Xception model based on the authors' implementations [8] of FGSM, MIFGSM, DIM, TIFGSM, TIFMIFGSM, and TIDIM. Hence, we leave these baselines' results empty for the Xception model. Similar to the results of Inc-v3, for the transferability results, our method, *i.e.*, ABBA$_{pixel}$, achieves slightly lower success rate than the state-of-the-art additive-perturbation-based attacks, *i.e.*, DIM and TIDIM, when attacking Inc-v3, Inc-v4, and IncRes-v2, and obtains higher success rate than TIDIM when attacking the Xception model. For the whitebox attacks, TIMIFGSM and MIFGSM usually achieve the highest success rate.

# 4 Hyper-parameter Analysis and Ablation Study

**Effect of $\epsilon$ and $\epsilon_\theta$.** We calculate the success rate of our method with different $\epsilon$ and $\epsilon_\theta$ in the Eq. (5) of our main paper, respectively. Specifically, we try $\epsilon$ with the range $[5, N]$ where $N = 51$ and $\epsilon_\theta$ in $[0, 1]$. As shown in Fig. III (a), the success rates become gradually higher with the increase of $\epsilon$ and

**Figure III.** Up: shows the success rate of AB₿A w.r.t. the variation of both $\epsilon$ and $\epsilon_\theta$ in Eq. (5) of our main paper where $\epsilon$ is within $[5, 50]$ with step size 5 and $\epsilon_\theta$ is in $[0, 1]$ with step size $0.1$. Down: shows an example of .

**Figure IV.** Up: two examples of AB₿A$_{pixel}$, AB₿A$_{obj}$, AB₿A$_{bg}$, AB₿A$_{image}$, and AB₿A. Bottom: Success rates of our method with respect to the object motion directions.

$\epsilon_\theta$. The highest success rates are $94.8\%$, $68.5\%$ $68.4\%$, and $72.1\%$ on Inc-v3, Inc-v4, IncRes-v2, and Xception, respectively. We also visualize adversarial examples of an image that has been successfully attacked on all $\epsilon > 0$ and $\epsilon_\theta > 0$. Obviously, as $\epsilon$ and $\epsilon_\theta$ increase, the visual effects of adversarial examples gradually become worse and the perturbations are more easily perceived. According to numerous attacking on different images, we choose the $\epsilon = 15.0$ and $\epsilon_\theta = 0.4$ to balance the success rate and visual effects when comparing with baselines on transferability in Sec. 3.2 in our main paper.

**Effect of motion directions.** we fix $\epsilon_\theta = 0.4$ and $\epsilon = 15.0$ and tune the motion direction of object and background by setting different x-axis and y-axis translations. For each object motion direction, we calculate the mean and standard variation of the success rates on different background moving directions. As shown in Fig. IV (B), the success rate increases as the object motion direction becomes larger in $[10°, 50°]$ while decreasing as the direction is smaller in $[50°, 70°]$. The success rate variation has symmetrical trend in the range of $[90°, 170°]$. Such results are mostly caused by the $L\infty$ used for constraining the translation. The motion direction is directly related to the translation and the success rate reaches the highest value around $45°$.

**Effect of blurred regions and importance of adaptive translations.** As reported in Tab. 1 in our main paper and cases shown in Fig. IV (U), AB₿A$_{pixel}$ achieves the highest attack success rate and transferability among all variants, which, however, changes the original image obviously and looks unnatural. AB₿A$_{obj}$ and AB₿A$_{bg}$ have the worst success rate on all models although they tend to generate visually natural motion blur. AB₿A$_{image}$ and AB₿A make good balance between the attack success rate and visual effects. In particular, AB₿A that jointly tunes the object and background translations can obtain much better transferability across normal trained and defense-based models. Note that, when compared with the results using fixed motion directions in Fig. IV (B), AB₿A obtains

**Figure V.** Left: the interpretable maps of six adversarial examples generated by FGSM, MIFGSM, and ABBA, respectively, with four models. Right: the transferability & consistency distributions of adversarial examples generated by the three attacks.

the highest success rate among all motion direction, further demonstrating usefulness of adaptive translations.

## 5 Interpretable Explanation of the Transferability

In the following, we explore the difference between ABBA, FGSM, and MIFGSM on the transferability. Note that, we implement FGSM and MIFGSM on the same platform (*i.e.*, pytorch with foolbox 2.3.0) with ABBA for fair comparison. We modify the method in [4] that generates an interpretable map for a classification model $f(\cdot)$ with a given perturbation. Then, we observe that the transferability of an adversarial example generated by an attack correlates with the consistency of interpretable maps of different models. Specifically, given an adversarial example $\mathbf{X}^{adv}$ generated by an attack and the original image $\mathbf{X}^{real}$, we can calculate an interpretable map $\mathbf{M}^{f}$ for $f(\cdot)$ by optimizing:

$$\underset{\mathbf{M}^{f}}{\arg\min} \quad f_y(\mathbf{M}^{f} \odot \mathbf{X}^{adv} + (1 - \mathbf{M}^{f}) \odot \mathbf{X}^{real}) + \lambda_1 \|\mathbf{M}^{f}\|_1 + \lambda_2 \mathrm{TV}(\mathbf{M}^{f}) \tag{1}$$

where $f_y(\cdot)$ denotes the score at label $y$ that is the ground truth label of $\mathbf{X}^{real}$ and $\mathrm{TV}(\cdot)$ is the total-variation norm. Intuitively, optimizing Eq. (1) is to find the region that causes misclassification. We optimize Eq. (1) via gradient decent in 150 iterations and fix $\lambda_1 = 0.05$ and $\lambda_2 = 0.2$. We can calculate four interpretable maps for each adversarial example based on four models, *i.e.*, Inc-v3, Inc-v4, IncRes-v2, and Xception, as shown in Fig. V(L). We observe that the interpretable maps of our method have similar distributions across the four models while the maps of FGSM and MIFGSM do not exhibit this phenomenon. To further validate this observation, we calculate the standard variation across the four maps at each pixel and get a value by mean pooling. We normalize the value and regard it as the consistency measure for the four maps. As shown in Fig. V(R), the consistency of adversarial examples of our method is generally higher than that of FGSM and MIFGSM. We further study the transferability of an adversarial example across models. Given an adversarial example from Inc-v3 and a model $f(\cdot)$, we calculate a score to measure the transferability under this model: $t^f = f_c(\mathbf{X}^{adv}) - f_y(\mathbf{X}^{adv})$ where $c \neq y$ is the label having maximum score among non-ground-truth labels. If $t^f > 0$ means the adversarial example fool $f(\cdot)$ successfully, and vice versa. As shown in Fig. V(R), the transferability of adversarial examples of our method is generally higher than that of FGSM and MIFGSM.

## 6 Attack results of STN-based model

As introduced in Sec. 2.3 in the main paper, we employ the spatial transformer network (STN) to tune the translation parameters of the object and background, and it may post a question if our method could also be useful in attacking STN-based CNN models. Towards more comprehensive evaluation and answering this question, we further conduct an experiment to attack the STN-based CNN implemented by [10] on the MNIST dataset. *First*,

**Table III.** Succ. rates of ABBA and NormalBlur before (Adv. from Inc-v3) and after deblurring via original and re-trained DeblurGANv2s.

| | Adv. from Inc-v3 | | | DeblurGANv2 | | | Re-DeblurGANv2 | | |
|---|---|---|---|---|---|---|---|---|---|
| | Inc-v3 | Inc-v4 | IncRes-v2 | Inc-v3 | Inc-v4 | IncRes-v2 | Inc-v3 | Inc-v4 | IncRes-v2 |
| ABBA | 65.3 | 31.1 | 30.0 | 31.4 | 24.2 | 18.4 | 22.9 | 22.5 | 16.8 |
| NormalBlur | 36.4 | 20.8 | 18.5 | 15.2 | 10.0 | 4.7 | 26.4 | 18.1 | 13.7 |

we use our ABBA to attack the STN-based CNN with different hyper-parameters, *i.e.*, the maximum translations ($\epsilon_\theta$) and the maximum number of valid kernel elements ($\epsilon_\theta$).

As shown in Fig. VI, the success rate of our method gradually increases as the $\epsilon_\theta$ and $\epsilon$ become larger, which demonstrates the effectiveness of our method. *Second*, we show several examples of our attack results in Fig. VIII with two groups of hyper-parameters, *i.e.*, $\epsilon_\theta = 0.5, \epsilon = 15.0$ and $\epsilon_\theta = 1.0, \epsilon = 15.0$. As shown in Fig. VIII, with the small $\epsilon_\theta = 0.5$, our method can generate slightly blurred handwritten digits that look naturally but fool the STN-based CNN. When using larger $\epsilon_\theta = 1.0$, the adversarial images become more blurred but still natural.

**Figure VI.** Success rate of ABBA w.r.t. different $\epsilon_\theta$ and $\epsilon$ that are maximum translations and maximum number of valid kernel elements.

# 7 Benefits to Blur-Robustness Enhancement

We conduct an experiment that trains the IncResv2 on the clean imagenet containing 1.1 million images and a modified imagenet with 1.3 million images including 0.2 million ABBA-blurred images, and evaluate the accuracy on the motion-blur subsets of ImageNetC [11]. We see that the Top-1 error of IncResv2 decreases from 73.0% to 53.2% with our blurred images, which strongly demonstrates the impact of ABBA to enhance the blur-robustness of DNNs.

# 8 More results of ABBA_physical

In Figure VII, we present four more examples generated by ABBA_physical that captures real-world images according the camera moving parameters estimated by ABBA. We try our best to compare synthesis images from ABBA with the physical images from ABBA_physical, demonstrating the motion blurred images generated by our method could be really found in the real world.

# 9 Defense Results of Re-trained DeblurGANv2 on ABBA and NormalBlur

NormalBlur generates motion-blurred image by optimizing Eq. (5) while fixing all kernel elements as $\frac{1}{N}$, which is equivalent to averaging neighbouring video frames where object and background move uniformly. In contrast, ABBA effectively tunes kernel elements to fool DNNs. Actually, the intention of Sec 3.5 is to study the effectiveness of existing deblurring method (*i.e.*, the 'already-deployed' deblurring modules) in defending the attack of ABBA with the tunable kernels. More detailed response: ❶ **NormalBlur** utilizes Eq. (5) to generate motion blur and **has considered background motion** via optimizing $\theta_b$. ❷ In practice, our assumption is that **we cannot get real motion information in the scene** and there is only one given static image. Hence, our attack is conducted under this assumption, *i.e.*, the object and background move uniformly (*i.e.*, at fixed speed) in a short time, which is a common phenomenon in the real world (*e.g.*, walking). Our attack could be easily extended to other cases where more motion information is available (*e.g.*, video). ❸ With the DeblurGANv2 trained on normal motion blur dataset (*e.g.*, GOPRO [22]), the decrease in succ. rate before and after deblurring in Table III have shown that the NormBlur can be defended more easily than ABBA, which demonstrates ABBA's tunable kernels facilitate achieving high attack success rate and anti-deblurring capability. **As suggested by the reviewer, when we further retrained the DeblurGANv2 with blurred images from ABBA.** ABBA can be defended more easily, which further indicates a promising direction of combining ABBA and the existing deblurring method for effective defense.

**Figure VII.** Comparing the visualization examples of ABBA$_{physical}$ with those of ABBA.

**Figure VIII.** Visualization examples of ABBA for attacking STN-based CNN with two group hyper-parameters, *i.e.*, $\epsilon_\theta = 0.5, \epsilon = 15.0$ and $\epsilon_\theta = 1.0, \epsilon = 15.0$.

## Footnotes

[1]https://github.com/tsingqguo/ABBA

[2]https://github.com/anlthms/nips-2017

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

| Original | ABBA$_{pixel}$ | ABBA | DIM | TIDIM |
|----------|----------------|------|-----|-------|

**Figure IX.** Seven visualization results of ABBA$_{pixel}$, ABBA, DIM, and TIDIM. All adversarial examples fool the Inc-v3 model.

**Figure X.** Seven visualization results of ABBA$_{pixel}$, ABBA, DIM, and TIDIM. All adversarial examples fool the Inc-v3 model.