[Reviews · NeurIPS 2020]

Review 1

Summary and Contributions: This paper presents a novel adversarial attack method based on motion blur. The method can generate visually natural motion-blurred images that can fool DNNs for visual recognition. Specifically, the paper first formulates kernel-based attack where each pixel has its own blur kernel. The adversarial blur kernels are then obtained by maximizing the loss function for visual recognition. Then, the paper formulates another attack that produces visually more natural images by constraining the blur kernel. Finally, the paper presents another variant whose blur kernels can be implemented in the real-world by actual camera and object motions. The experimental results show that the proposed methods can generate successful attacks and the possibility of the attack using real-world adversarial motion blur.

Strengths: The adversarial attack is an interesting topic that recently has attracted much attention as it exposes the weakness of DNNs. While several kinds of attacks have recently been presented, as far as I understand, this paper is the first one that shows the possibility of the blur based adversarial attack, which can spark many following pieces of research. The paper is well written, and the proposed methods are convincing. The formulations presented in the paper are intuitive and compelling. The results also look natural and convincing. Especially, the study of the real-world blur attack is interesting as it shows the possibility of physical attacks.

Weaknesses: It would be much more interesting if the proposed method were able to fool DNNs in such a way that the fooled DNNs recognize input images as some target objects that the attacker wants them to identify. The paper presents only one real-world example, which is far from enough to prove the possibility of the real-world adversarial attack.

Correctness: I think so.

Clarity: The paper is well written. Typos & grammatical errors: • 5: great important ==> greatly important, or just important • 34: naturally found `naturally' ==> found `naturally' ? • 210: following ==> follow • 215: 1) Capturing a picture ... . 2) Using ... , ==> 1) capturing a picture containing an object in a real-world scene, 2) using ... , 3) ... , and (4) moving ... • 221: primary ==> primarily • 224: 5 research questions ==> five research questions • 224: 1): Is the ==> 1) Is the • 227: 4): ==> 4) • 256: images. E.g., ==> images, e.g.,

Relation to Prior Work: Yes

Reproducibility: Yes

Additional Feedback: == After rebuttal == After reading the rebuttal and other reviews, I keep my original score. As far as I know, this paper is the first paper that shows the possibility of blur-based adversarial attack, which I believe is interesting enough to the computer vision community. While I am not very familiar with this topic, in my opinion, this kind of adversarial attack is worthy as long as it can deceive target systems and defenders who are not prepared for this kind of attack. From this perspective, I think all the results (except for the results of ABBA_pixel) look reasonably natural enough to deceive unprepared defenders. I have a doubt about its effectiveness to deceive the system as I suspect that simply decreasing the accuracy of the visual recognition system can be really practical as an attack. However, the rebuttal also partly shows a targeted attack can be possible, so I am satisfied with the paper, and believe that more effective blur-based attacks can be further investigated by future works.


Review 2

Summary and Contributions: The paper comprehensively investigate the potential hazards of blur effect for DNN and propose a novel adversarial attack method that can generate visually natural motion-blurred adversarial examples, named motion-based adversarial blur attack (ABBA).

Strengths: (1) To the best of my knowledge, this is the first paper to propose a adversarial attack method that can generate motion/kernel blurred based adversarial examples. (2) The paper also introduce a saliency regularizer that forces consistent blur patterns within the boundary of the objects to generate realistic blur.

Weaknesses: (1) In Section 3.5, the author claimed that "The 'normal' motion blur synthesized by averaging neighbouring video frames is equivalent to set the kernel weights as 1/N to average the translated examples Eq. (4)", however i respectively disagree on that (a) we cannot assume the background is totally static across multi frames, the saliency based motion generation methods only apply motion on foreground object, thus ignoring any motion in the background (b) The Eq. (4) also assume object is moving in a static speed, which is also not true in motion generated by averaging methods. (c) In the supplementary Figure IV, we can also see a clear difference between Adversarial-Blurred Images and Real blurred Image. There is clearly a domain gap between 'normal' motion blur and 'adversarial' motion blur. Thus i think experiment of SOTA deblurring methods in Sec. 3.5 should be redone on a fair setting where SOTA deblurring methods is also trained on 'adversarial' motion blur. (2) The adversarial motion blur is still limited since it's still using kernel-based methods to generate blur, which is less realistic than 'normal' methods by averaging multiple frames. (3) Some minor typos: Figure 2: orignal -> original

Correctness: I have some doubt one some of the claims, see Weakness.

Clarity: The paper is well-written and easy to understand

Relation to Prior Work: Yes

Reproducibility: Yes

Additional Feedback:


Review 3

Summary and Contributions: This paper proposed a new method to create adversarial examples by blurring samples. Experimental studies show somehow its effectiveness.

Strengths: The paper is well represented. Idea is interesting. The coverage of the experimental study seems broad.

Weaknesses: 1, The idea is interesting, while it seems the idea of creating adversarial examples by blurring has been studies before, e.g., the compared baselines in the experimental study. 2, For comparison, there is a baseline method of blurring by averaging continuous frames. This one should be compared. 3, The related work is shallow. Some methods about blurring should be discussed. There are work about blurring for deblurring. 4, The results in Table 1 are not strong, especially in the first part (the attacking results). 5, It is not reasonable that the ABA_pixel is better than ABA in the Table 1. Can the authors explain it? 6, Minor: Tab. 3.2 should be Tab. 1.

Correctness: See the weakness please.

Clarity: Yes.

Relation to Prior Work: Not sufficient. More related work should be discussed.

Reproducibility: Yes

Additional Feedback:


Review 4

Summary and Contributions: This paper proposes a novel adversarial attack method to produce motion-blurred adversarial examples. By tuning the blur-kernels weights to the clear input image, the proposed method further improves the visual motion-blurred effects by regularizing the blur kernel of salient object in the image.

Strengths: As the author said, this is the first attempt to investigate kernel-based adversarial attacks. The method seems sound.

Weaknesses: Even though the author claims that the proposed method is able to generate adversarial images of more plausible appearance, compared with other noise-based methods, I don’t think motion blur is a good choice for the adversarial attack algorithms. Motion blurs are more notable in the images and easier to detect in the input compared with the noise-based attacks. The goal for generating adversarial images is to improve the classifier’s performance when encounter two images having similar high-level features or visually the same. However, the introducing of motion blur can change the global consistency among the high-level features of classifier. The author states that the proposed motion blur attacks are hard to remove by deblurring methods than normal motion blurs, which in my opinion, doesn’t make any sense. Based on the results and how the motion blur is constructed in this paper, the synthesized blurs are more likely to be applied on the whole image, instead of on a specific object (Figure 2 and 5). Obviously, the proposed results failed at resembling the real motion blur on plenty of cases. Essential experimental comparisons are missed. I will suggest the authors to compare with other learning-based method to synthesize motion blurs (for example [14]). -----------------AFTER REBUTTAL--------------------------------------- After carefully reading other reviewers' comments and the author's rebuttal feedback, I decided to stick to my original grade opinion. I appreciate the efforts the authors have devoted to the paper. However, I am still not very convinced by the novelty of using motion-blur (or any kind of blur) as an adversarial attack algorithm for training a good classifier. The noise-based adversarial attack methods provide insights on DNN training, as they carefully integrate the noise into the images and make the added noise hard to detect and remove. In contrast, due to the loss of the image's high-frequency content, one can easily filter out the blur-attacked images, containing obvious/large blurriness, using a blur-detector. As for the images with subtle/small motion blur, a regular deblurring algorithm can fail the proposed attacks, as demonstrated in Figure 6 (even the normal motion-blur algorithm by averaging neighboring frames outperforms the proposed one among small blur kernel range). It is true that the proposed attack can deceive the target systems and defenders without preparation for the blur-attack. Nevertheless, as this kind of attack is easy to detect (for large blur) and remove (for small blur), I doubt the practicability of the proposed method.

Correctness: Yes

Clarity: Readable, but not well organized

Relation to Prior Work: Yes

Reproducibility: Yes

Additional Feedback:

[Author Response · NeurIPS 2020]

**Common response:** We re-emphasize our key contribution and novelty: ❶ **Difference from adversarial noise and other blurs.** The motion blur and additive noise are 2 levels/scopes of perturbations that are orthogonal from the perspective of camera perception. Specifically, additive noise mainly investigates the influence of additive distortions on the received image to DNNs, while motion blur considers the perception system's front-end, *i.e.*, the motion of object or camera. Motion blur often occurs in the physical process of practical image perception and can potentially post serious effects on safety and security, making it of great importance.

Compared with other image blurs (*e.g.*, defocus blur), motion blur, as an intrinsic phenomenon, directly relates to the motion of object and camera and cannot easily be removed by adjusting camera setting. ❷ **Key contribution.** Although extensive work has been conducted on attacking/defensing for adversarial noise, up to present, limited studies have been performed on how motion blur affects DNN-based prediction. This work initiates the first step to comprehensively investigate motion blur effects of camera perception from the perspective of adversarial attack and proposes the motion-based adversarial blur attack (ABBA). ❸

| | Adv. from Inc-v3 | | | DeblurGANv2 | | | Re-DeblurGANv2 | | |
|---|---|---|---|---|---|---|---|---|---|
| | Inc-v3 | Inc-v4 | IncRes-v2 | Inc-v3 | Inc-v4 | IncRes-v2 | Inc-v3 | Inc-v4 | IncRes-v2 |
| ABBA | 65.3 | 31.1 | 30.0 | 31.4 | 24.2 | 18.4 | 22.9 | 22.5 | 16.8 |
| NormalBlur | 36.4 | 20.8 | 18.5 | 15.2 | 10.0 | 4.7 | 26.4 | 18.1 | 13.7 |

**Figure R-1:** (Top-L) subfigure shows two examples of the targeted attack via ABBA. (Top-R) subfigure shows four examples of our ABBA and ABBA$_{Pyhsical}$ that performs attack in the real-world with the estimated translation parameters of ABBA. (Bottom) Succ. Rate of ABBA and NormalBlur before (Adv. from Inc-v3) and after deblurring via existing and retrained DeblurGANv2s.

**Benefits to blur-robustness enhancement (R4).** We conduct an experiment that trains the IncResv2 on the clean imagenet (1.1M) and a modified imagenet (1.3M) containing ABBA-blurred images (0.2M), and evaluate the accuracy on the motion-blur subsets of ImageNetC. We see that the Top1 error of IncResv2 decreases from 73.0% to 53.2% with our blurred images, which strongly demonstrates the impact of ABBA to enhance the blur-robustness of DNNs.

**Q1 (R1):** **Targeted attack (TA) and more real-world examples.** We can intuitively achieve the **TA** that is to fool a classifier to predict a specified category by replacing the max objective function (Eq. (5)) with a min objective function towards the specified category. We give two TA examples and our real-world examples in Fig. R-1(T).

**Q2 (R2):** **Explanation of the NormalBlur in Sec. 3.5.** NormalBlur generates motion-blurred image by optimizing Eq. (5) while fixing all kernel elements as $\frac{1}{N}$, which is equivalent to averaging neighbouring video frames where object and background move uniformly. In contrast, ABBA effectively tunes kernel elements to fool DNNs. Actually, the intention of Sec 3.5 is to study the effectiveness of existing deblurring method (*i.e.*, the 'already-deployed' deblurring modules) in defending the attack of ABBA with the tunable kernels. We thank the reviewer's suggestion in using the deblurring modules trained from ABBA. More detailed response: ❶ **NormalBlur** utilizes Eq. (5) to generate motion blur and **has considered background motion** via optimizing $\theta_b$. ❷ In practice, our assumption is that **we cannot get real motion information in the scene** and there is only one given static image. Hence, our attack is conducted under this assumption, *i.e.*, the object and background move uniformly (*i.e.*, at fixed speed) in a short time, which is a common phenomenon in the real world (*e.g.*, walking). Our attack could be easily extended to other cases where more motion information is available (*e.g.*, video). ❸ With the DeblurGANv2 trained on normal motion blur dataset (*e.g.*, GOPRO [22]), the decrease in Succ. Rate before and after deblurring in Fig. R-1(B) have shown that the NormBlur can be defended more easily than ABBA, which demonstrates ABBA's tunable kernels facilitate achieving high attack success rate and anti-deblurring capability. **As suggested by the reviewer, when we further retrained the DeblurGANv2 with blurred images from ABBA.** ABBA can be defended more easily, which further indicates a promising direction of combining ABBA and the existing deblurring method for effective defense.

**Q3 (R4):** **ABBA does not generate the motion blur on the whole image.** As defined in Eq. (4) and (5), ABBA jointly (but differently) tunes the object and background's translation parameters (*i.e.*, $\theta_o$ and $\theta_b$) to generate motion-blurred adversarial images. The visualization results in Fig. 3 in the submission, Fig. III and Fig. VII in the supplementary material all already demonstrate that the motion blur of object and background can be different.

**Q4 (R3):** **Explanation of ABBA's performance. ❶ ABBA$_{pixel}$ vs. ABBA.** ABBA$_{pixel}$ achieves strong attack capability since we perform fine-grained tuning for the kernel of each pixel independently (Eq. (3)). However, this can make ABBA$_{pixel}$ generate perceptible noise-like images (Fig. 2). To generate more realistic blur, we further propose the ABBA that uses the saliency regularization to constraint kernels to be the same in both object and background regions, which trades off the attack success rate a bit. ❷ **Advantages over averaging neighbouring video frames and SOTA noise-based attacks.** We have already compared ABBA with the 'averaging neighbouring video frames', *i.e.*, NormalBlur in Sec. 3.5 and the column 'Adv. from Inc-v3' of Fig. R-1(B). Obviously, ABBA achieves much higher success rate and transferability than NormalBlur. Moreover, compared with SOTA noise-based attacks (Tab. 1 in submission), ABBA$_{pixel}$ and ABBA obtain the best and second best results across all defense methods.

**Q5 (R4):** *c.f*. **[14].** We have made our best efforts to cite and compare with [14]. The failure for experimental comparison is due to missing of key data/model components and fundamentally technical differences: ❶ We have made private communication with the authors [14] for pre-trained models and training data. However, both are unavailable due to commercial reasons. ❷ Technically, [14] needs two neighboring frames as the input while we focus on generating visually natural motion blur with only one static image as inputs. Moreover, [14] relies on an offline-trained UNet for realistic motion blur instead of and without focusing on conducting the attack.

[Meta-Review · NeurIPS 2020]

This paper presents a novel adversarial attack method based on motion blur. The method can generate visually natural motion-blurred images that can fool DNNs for visual recognition. The paper is well written, and the proposed methods are convincing. One reviewer is convinced on the goodness of the paper and suggest a clear acceptance. A second and a third ones consider the paper above the acceptance threshold, being the problem very interesting and the approach clear. this is the first attempt to investigate kernel-based adversarial attacks. A fourth reviewer, appreciated the paper and the rebuttal but did not changed the idea on the fact that the paper is below a threshold of acceptance, mainly because, he/she said, a blur-detector could be used to improve the image and thus avoid the attack. After a long discussion, the consensus has not reached but the majority of reviwers agree in the acceptance. Also the AC agrees.